# Lens fluorescence and skin fluorescence in the Copenhagen Twin Cohort Eye Study: Covariates and heritability

**Jakob Bjerager** [1]*, **Sami Dabbah**[1], **Mohamed Belmouhand**[1], **Simon P. Rothenbuehler**[1,2], **Birgit Sander**[1], **Michael Larsen**[1,3]

**1** Department of Ophthalmology, Rigshospitalet, Glostrup, Denmark, **2** Department of Ophthalmology, University Hospital Basel, Basel, Switzerland, **3** Faculty of Health and Medical Sciences, Department of Clinical Medicine, University of Copenhagen, Copenhagen, Denmark

* jakob.kondras.bjerager.02@regionh.dk

**Data Availability Statement:** Data is publicly available through Dryad: https://doi.org/10.5061/dryad.rxwdbrv7w.

## Abstract

Lens and skin fluorescence are related to the systemic accumulation of advanced glycation end products, which is accelerated in diabetes. We have examined lens fluorescence and skin fluorescence in healthy adult twins. The study enrolled twins aged median 59 years from a national population-based registry. Diabetic individuals were excluded from analysis. The interrelatedness between fluorescence parameters and relations between fluorescence and age, current $HbA_{1c}$ and smoking pack years were examined using correlation tests and mixed model linear regression analyses. Broad-sense heritability was analyzed and compared for lens fluorescence, skin fluorescence and $HbA_{1c}$. Lens fluorescence and skin fluorescence were crudely interrelated (R = 0.38). In linear regression analyses, age explained a larger fraction of the variance in lens fluorescence ($R^2$ = 32%) than in skin fluorescence ($R^2$ = 20%), whereas $HbA_{1c}$ explained smaller variance fractions ($R^2$ = 3% and 8%, respectively) followed by smoking pack years (4% and 3%, respectively). In multivariate analyses, age, $HbA_{1c}$ and smoking pack years combined explained more of the variance in lens fluorescence ($R^2$ = 35%) than in skin fluorescence ($R^2$ = 21%), but the influence of $HbA_{1c}$ on lens fluorescence was not statistically significant (p = .2). Age-adjusted broad-sense heritability was 85% for lens fluorescence, 53% for skin fluorescence and 71% for $HbA_{1c}$ in best fitting heritability models. Both fluorescence parameters increased with age, current glycemia and cumulative smoking. Lens fluorescence was found to be a predominantly heritable trait, whereas skin fluorescence was more influenced by environmental factors and closer related to current glycemia. The results suggest that skin fluorophores have a faster turnover than lens fluorophores.

## Introduction

The fluorescence of various tissues with high proportions of long-lived proteins increase with age as the result of denaturation of various constituent molecules and accumulation of

**Funding:** This work was supported by THE VELUX FOUNDATIONS (JB, grant no. 00028975, https://veluxfoundations.dk/en), Rigshospitalets Forskningsudvalg (MB, grant no. E-23334-02, https://www.rigshospitalet.dk/forskning/om-forskningen/Sider/forskningsudvalget.aspx), P. Carl Petersens Fond (MB, grant no. 19102, https://www.pcarlp-fond.dk/), Helsefonden (MB, grant no. 19-B-0063, https://helsefonden.dk/), Aase og Ejnar Danielsens Fond (MB, grant no. 18-10-0698, https://danielsensfond.dk/), Beckett Fonden (MB, grant no. 19-2-3490, https://beckett-fonden.dk/) and Einar Willumsens Mindelegat (MB, grant no. 500028, https://www.legatbogen.dk/fabrikant-einar-willumsens-mindelegat/stoetteomraade/7684). This work was also supported by Horizon 2020, the European Union's Framework Programme for Research and Innovation, under grant agreements no. 732613 (GALAHAD) and no. 780989 (MERLIN). The funding organization had no role in the design or conduct of this research.

**Competing interests:** The authors have declared that no competing interests exist.

degradation products [1]. In the lens and skin, the most prominent contribution to the formation of fluorophores stems from spontaneous non-enzymatical reactions between sugars in the bodily fluids and amino groups of proteins, which result in the formation of advanced glycation end products (AGEs) [2–10]. Numerous studies have found lens fluorescence (LF) and skin fluorescence (SF) to be elevated in diabetes and in long-term diabetes complications [11–25]. Hemoglobin $A_{1c}$ ($HbA_{1c}$) levels, conventionally used for diagnosing and monitoring type 2 diabetes, are altered by genetics, erythrocyte pathology, certain medications as well as kidney and liver disease [26–28] and the biomarker is unable to capture short-term yet metabolically stressful hyperglycemic spikes [29–32]. Cross-sectional $HbA_{1c}$ only explains a small fraction of the risk of diabetes complications, as for example 11% of the risk of developing diabetic retinopathy [33]. The duration and past severity of diabetes at the time of diagnosis can only be inferred from the presence of diabetic complications and hence there is a need for a palette of biomarkers that can combine to provide a better understanding of the maintenance condition of a given individual [30,34]. Quantitative tissue fluorometry has been suggested as potential tools for non-invasive diabetes screening, for assessment of cumulative glycemic exposure and for complication risk assessment in diabetes [34,35], but applicability is challenged by confounding factors also affecting fluorescence, including age, smoking and genetics [13,15,19,36–42].

Research into the interrelatedness and normal variation of tissue fluorescence parameters may broaden our understanding of tissue ageing kinetics as well as assist in evaluating the clinical potential of these methods in management of diabetes, the metabolic syndrome and cardiovascular disease, all of which are related to increased tissue fluorescence [12,13,23,41,43,44]. We have conducted a comparative study of lens and skin fluorescence and examined their relationships to covariates in healthy adult twins.

## Methods

### Setting and study population

This study was a part of the 2019–20 round of the Copenhagen Twin Cohort Eye Study, which was conducted by a clinical ophthalmology department (Dept. of Ophthalmology, Rigshospitalet, Glostrup, Denmark) in collaboration with a national twin registry (the Danish Twin Register, University of Southern Denmark). Female and male same-sex twin pairs were invited to participate. Inclusion criteria for participation were age >18 years and signed informed consent. Exclusion criteria were conditions that precluded imaging of the fundus of the eye, uncontrolled glaucoma with ocular tonometry >30 mmHg and cognitive impairment that hindered informed consent. Pupil dilation by mydriatics was made only in eyes with intraocular pressure lower than 22 mmHg (iCare TA01i, Icare Finland Oy) and a central anterior chamber depth ≥2.3 mm (IOLMaster 700, Carl Zeiss Meditech AG) or a van Herrick anterior chamber angle class 2 or higher. The study was conducted in accordance with the Declaration of Helsinki after approval from the national medical ethics committee (National Videnskabsetisk Komité, approval no. H-18052822). Lens and skin fluorometry was planned and attempted in all individuals who were examined in 2019, limitations being late instrument delivery and expiring instrument lease contract of the skin fluorometer. Subjects with diabetes were excluded due to strong outlier effects. The prevalence of subjects with diabetes was too low to allow for meaningful comparisons between subjects with and without diabetes. Diabetes was defined as $HbA_{1c}$ ≥48 mmol/mol at the study visit or a medical history of type 2 diabetes. No individuals had type 1 diabetes. Heritability analyses for $HbA_{1c}$ were based on all non-diabetic twin pairs included in the cohort study.

## Data sources

Lens fluorescence was measured in the right eye of phakic subjects using a commercial ocular fluorometer (Fluortron Master TM-2 with Windows software, revision B.17, OcuMetrics, Mountain View, California, USA), approximately 1 hour after dilation with tropicamide 1% eye drops. The device measures blue-green fluorescence at incremental steps of 0.125 mm along the optical axis of the eye using excitation light at 430–490 nm and detection at 530–630 nm with results reported in units of equivalent fluorescein concentration in water (ng/mL). Measurements were performed under scotopic lighting conditions. Absorption-corrected anterior lens peak fluorescence was calculated using the manufacturer's software. Lens fluorescence peak values were corrected for ambient light in the examination room by subtraction of the lowest fluorescence readings in each scan (averaged from the fluorescence intensities of the 15/148 steps with the lowest intensities). Subjects were scanned up to six times in order to achieve three successful scans. Individuals were excluded from the study if six scan attempts could not provide three successful scans. Unsuccessful scans counted scans with ambient background light values above 30% of the posterior absorption-corrected lens peak fluorescence, as recommended by the manufacturer, and if blinking had occurred at critical points during the scan. Lens fluorescence values used for data analysis were based on the average absorption-corrected anterior peak value of three successful scans. Study subjects with intraocular lens implants were excluded.

Skin fluorescence was measured on the anterior forearm with a designated commercial device (Diagnoptics AGE Reader, Diagnoptics Technologies B.V., Groningen, Netherlands). The instrument emits light at 300–420 nm, with peak intensity at 370 nm, on a 4 $cm^2$ skin area and measures emission at 300–600 nm. Data output is a double-digit arbitrary unit (AU) index of 420–600 nm fluorescence relative to reflected 300–420 nm emission multiplied by 100 [6]. The average of three readings per subject was used for analyses. According to the manufacturer's guidelines, subjects with excessive sweating, tattoos, recently applied skin cream or recent intensive sunbathing affecting the skin region of interest were excluded from analysis. The first measurement after device start-up was routinely discarded, as recommended by the manufacturer.

We choose to include three lens fluorescence and three skin fluorescence readings per subject for analyses since three are enough to evaluate reproducibility of measurements and the maximum to which one can reasonably expose a study participant in a study that also includes other procedures. Also, unilateral measurements only do not require statistical adjustments for paired organ data clustering.

Blood samples obtained during the examinations were analyzed for $HbA_{1c}$.

Data on accumulated smoking pack years were obtained by interview. Participants who reported <1 pack year were categorized as non-smokers.

## Outcome measures and covariates

Lens and skin fluorescence data were tested for interrelatedness and relation to age, $HbA_{1c}$ and smoking pack years using correlation tests and linear regression analyses. Broad-sense heritability analyses were conducted for fluorescence parameters and $HbA_{1c}$ with outcome variables being the heritability coefficients A (additive genetics, often referred to as $a^2$), D (dominant genetics, $d^2$), C (shared environment, $c^2$) and E (non-shared environment, $e^2$) and the broad-sense heritability coefficient $h^2$ (A + D). Heritability coefficients were examined in the following combinatory heritability models: ACE, ADE, AE, DE and CE.

## Statistical analyses

Microsoft Excel 360 for Windows 10 was used for demographic statistics, GraphPad Prism v9.0.0.121 for reproducibility of measurements analyses and R-Studio v1.2.5001 for Windows

10 was used for all other statistical analyses. Normality was tested by Shapiro-Wilk normality tests. Fluorescence parameters were transformed by log10 to obtain normal distributions in parametric tests. All fluorescence values reported have been back-transformed to geometric mean values with 95% confidence intervals. Parametric parameters were reported in means and standard deviations (SD) while non-parametric parameters were reported in medians and inter-quantile ranges (IQR). Reproducibility of fluorescence measurements were assessed by three-group one-way ANOVA analyses. Pearson's correlation tests were used in case of normal distributions and Spearman's rank correlation tests were used for non-normally distributed parameters. Univariate and multivariate log-level linear mixed model regression analyses adjusted for twin-pair clustering were performed with the R functions 'lmer()' (lme4 v.1.1.26 package) and 'modelTest ()' (JWileymisc v. 1.2.0 package). Reported coefficient estimates from linear regression analyses were transformed by antilog to designate percentage increase in fluorescence per unit increase in either age (years), smoking (pack years) or $HbA_{1c}$ (mmol/mol),. Broad-sense heritability was calculated for lens fluorescence, skin fluorescence and $HbA_{1c}$ by a linear regression model of each parameter as a function of age using the R function 'twinlm()' (mets v. 1.2.8.1 package). Best fitting heritability models were found by Akaike's information criterion (AIC). The lowest AIC-value defined the best fitting model for lens fluorescence, skin fluorescence and $HbA_{1c}$, but models with AIC-values between the value of the best fitting model and the value of the best model plus two AIC-units were considered non-inferior to true best fitting models.

## Results

This study included 239 subjects with lens fluorescence measurements and 177 individuals with skin fluorescence measurements (Fig 1, for demographics see Table 1). 163 individuals provided both lens and skin fluorescence data (median age 59 (IQR 13), median $HbA_{1c}$ 37 (IQR 4)). 318 non-diabetic paired twins from the cohort study (85 monozygotic and 74 dizygotic pairs) presented $HbA_{1c}$ data (Median age 60 (IQR 12), median $HbA_{1c}$ 37 mmol/mol (IQR 4)).

Mean lens fluorescence averaged from three measurements per subject was 554 ng/mL [95% CI 530–579] (n = 239). Mean skin fluorescence averaged from three measurements per subject was 2.05 AU [95% CI 1.99–2.11] (n = 177). We found both measurements of lens and skin fluorescence to have a high repeatability (S1 File).

Linear mixed model regression analysis of skin fluorescence as a function of lens fluorescence adjusted for twin-pair clustering found a marginal, positive relationship (adjusted $R^2$ = 0.15 (p < .001)) (Fig 2). In correlation testing, the correlation between lens fluorescence and skin fluorescence was R = 0.38 (p < .001). Positive correlations were found between fluorescence parameters and examined covariates (Table 2).

In univariate linear regression analyses, age was the parameter that explained the most variance in lens fluorescence and skin fluorescence (31.9% and 15.4% respectively based on $R^2$-values, both p < .001)). The multivariate regressions incorporating both age, $HbA_{1c}$ and smoking pack years were found to explain 35.0% and 20.8% of the variances in lens fluorescence and skin fluorescence, respectively. The influence of all covariates where statistically significant (all p≤.036), except for $HbA_{1c}$ in multivariate analysis of lens fluorescence (p = .204) (Table 3).

### Heritability of fluorescence parameters and $HbA_{1c}$

Broad-sense heritability analyses showed that the best fitting heritability model for lens fluorescence was the AE-model ($h^2$ = 85%), although the ACE-, ADE- and DE-models were all statistically non-inferior according to AIC criteria. The best fitting skin fluorescence model was

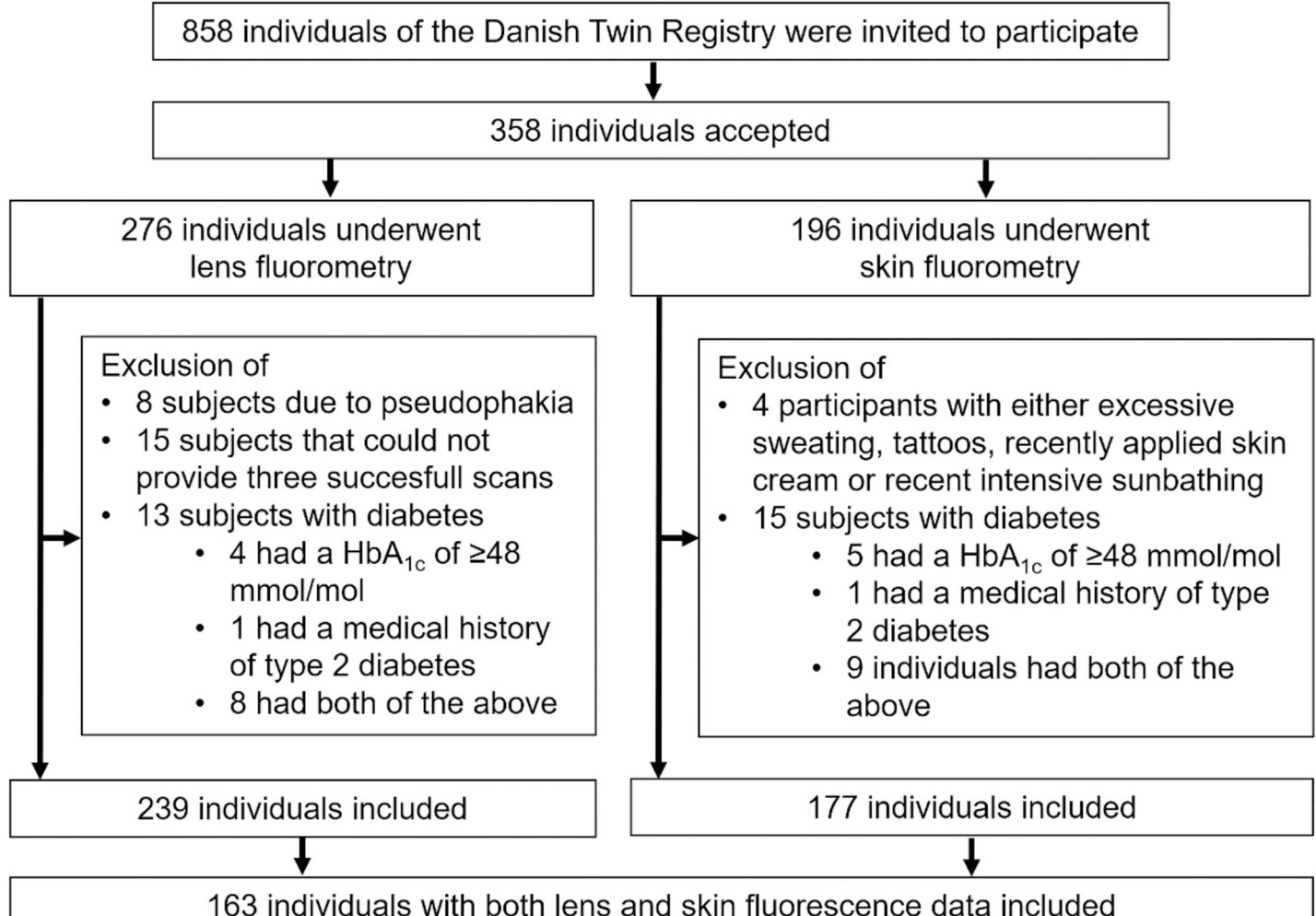

**Fig 1. Recruitment, exclusion and inclusion of study participants for analyses of lens and skin fluorescence.**

the DE-model ($h^2$ = 53%), with the ADE- and AE-models being statistically non-inferior (Table 4). For $HbA_{1c}$, the best fitting heritability model was the AE-model (AIC:1554.847) with the ACE- and ADE-models being non-inferior (AIC: 1555.083 and 1556.847, respectively).

Broad-sense heritability ($h^2$) of best fitting models was 85% for lens fluorescence, 53% for skin fluorescence and 71% for $HbA_{1c}$. Lens fluorescence had the highest $h^2$ across all models, skin fluorescence had the lowest $h^2$ in 3/4 models whereas $HbA_{1c}$ ranked in the middle between lens and skin fluorescence in 3/4 models. The exceptions to the trend were the ACE-models, where $HbA_{1c}$ had the lowest $h^2$ and skin fluorescence ranked in-between lens fluorescence and $HbA_{1c}$ (Table 4, Fig 3).

## Discussion

In our middle-aged, non-diabetic study population, age was the co-variate studied that explained the largest proportion of variance in both lens and skin fluorescence, and the influence was stronger on lens fluorescence. Lesser roles, statistically, were found for $HbA_{1c}$ and accumulated smoking. Combined, these covariates could only explain modest amounts of the

**Table 1. Demographics of lens and skin fluorescence study populations.**

| | Total population | Paired MZ twins | Paired DZ twins |
|---|---|---|---|
| **Lens fluorescence** | | | |
| **n** | 239 | 108 (54 pairs) | 100 (50 pairs) |
| **Lens fluorescence, ng/mL, [95% CI]** | 554 [530–579] | 536 [499–575] | 576 [540–614] |
| **Sex (females)** | 55% | 59% | 46% |
| **Age, years, (IQR)** | 59 (11) | 58 (12) | 60 (12) |
| **HbA$_{1c}$, mmol/mol, (IQR)** | 37 (4) | 37 (4) | 37 (4) |
| **Smokers, "yes" or "previous" (%)** | 45% | 47% | 41% |
| **Smoking pack years if smoking "yes" or "previous", (IQR)** | 11 (13) | 11 (12) | 11 (15) |
| **Skin fluorescence** | | | |
| **n** | 177 | 72 (36 pairs) | 86 (43 pairs) |
| **Skin fluorescence, AU, [95% CI]** | 2.05 [1.99–2.11] | 2.02 [1.92–2.12] | 2.04 [1.95–2.13] |
| **Sex (females)** | 49% | 53% | 44% |
| **Age, years, (IQR)** | 59 (14) | 58 (16) | 60 (14) |
| **HbA$_{1c}$, mmol/mol, (IQR)** | 37 (4) | 37 (4) | 37 (4) |
| **Smokers, "yes" or "previous" (%)** | 44% | 42% | 52% |
| **Smoking pack years if smoking "yes" or "previous", (IQR)** | 12 (13) | 11 (12) | 11 (15) |

There was no statistically significant difference between MZ and DZ subjects for any study parameters (all p>.05, determined by Mann-Whitney U tests). **AU:** Artificial Units, **CI:** Confidence interval, **DZ:** Dizygotic, **IQR:** Inter-quantile range, **MZ:** Monozygotic, **n:** Number of subjects.

variation in lens and skin fluorescence. As our fluorescence measurements were highly repeatable with no statistically significant difference between measurements of individuals, the considerable residual variance in fluorescence parameters left unexplained cannot be meaningfully attributed to a lack of precision in the fluorometric methods used, and it is likely better explained by genetical influences or environmental factors unaccounted for in the present study.

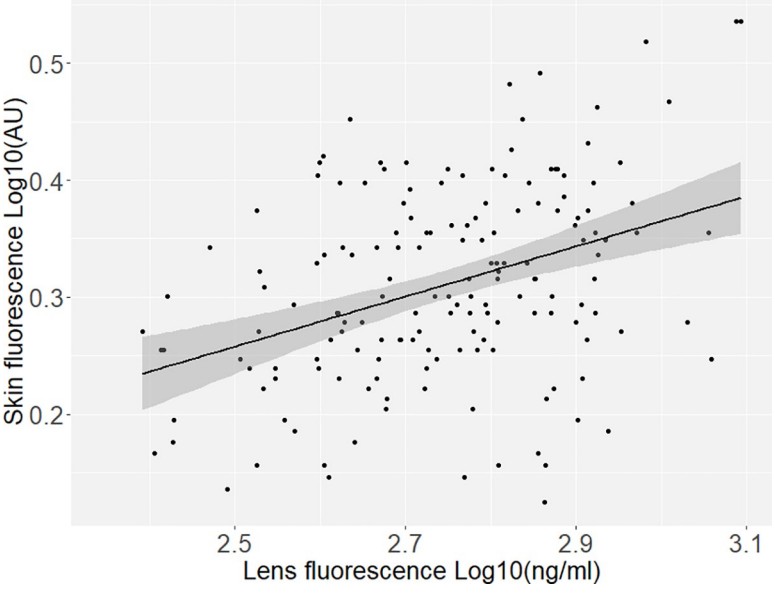

**Fig 2. The relationship between lens fluorescence and skin fluorescence (n = 163).** Fluorescence values was transformed by log10. Linear regression line with 95% confidence intervals shown. AU: Artificial Units.

**Table 2. Correlations between study parameters.**

|  | Correlation | p-value |
|---|---|---|
| **LF (n = 239)** |  |  |
| **Age** | 0.55 [S] | p < .001 |
| **HbA$_{1c}$** | 0.21 [S] | p = .001 |
| **Smoking** | 0.11 [S] | p = .100 |
| **SF (n = 177)** |  |  |
| **Age** | 0.35 [S] | p < .001 |
| **HbA$_{1c}$** | 0.31 [S] | p < .001 |
| **Smoking** | 0.15[S] | p = .048 |
| **LF$_{Log10}$ and SF$_{Log10}$ (n = 163)** | 0.38 [P] | p < .001 |

**DZ:** Dizygotic, **LF:** Lens fluorescence, **MZ:** Monozygotic

[S]: Spearman rank correlation

[P]: Pearson correlation, **SF:** Skin fluorescence.

As would be expected, the more long-term glycemia indices that are embedded in the lens and skin fluorophore concentrations correlated better with each other than with HbA1$_{1c}$, as the latter marker of glycemia-related protein denaturation reflects glycemia levels over no more than approximately 90 day. Cross-sectional glycemia levels may change considerably over a lifetime. Another factor to consider is that HbA$_{1c}$ is formed as early as in the second stage of the common chain of biochemical reactions that produce glycemia-related protein denaturation, whereas fluorescent AGEs that accumulate in long-lived tissues are formed with higher latency [45]. HbA$_{1c}$ therefor both concentrate and degenerate more rapidly than tissue-accumulated fluorescent AGEs.

We found correlations between lens fluorescence, skin fluorescence and covariates to be considerably lower than those found in a previous comparative study of lens and skin fluorescence among non-diabetic individuals by Januszewski et al. (e.g. correlations between lens and skin fluorescence of R = 0.38 compared to R = 0.58) [34]. Minor methodological discrepancies between the two studies may have contributed to differences in findings. More importantly, differences in sample sizes (n = 163 in the present study compared to n = 60 in Januszewski et al) and age groups (median age 59 years (IQR 11–14) in the present study compared to mean age 36 years ± SD 13 in the former study) may explain why the curve fit was more convincingly achieved by the former group of investigators, since the inter-individual variation in

**Table 3. Multivariate and univariate mixed model linear regression analyses.**

|  | Univariate analyses | | | Multivariate analyses | | |
|---|---|---|---|---|---|---|
|  | Estimate (%) | p-value | R$^2$ (%) | Estimate (%) | p-value | R$^2$ (%) |
| **Lens fluorescence$_{Log10}$ (n = 239)** |  |  |  |  |  |  |
| **Age, years** | 2.4 | p < .001 | 31.9 | 2.3 | p < .001 | 35.0 |
| **HbA$_{1c}$, mmol/mol** | 2.0 | p = .002 | 3.3 | 0.7 | p = .204 | - |
| **Smoking, pack years** | 0.6 | p < .001 | 4.1 | 0.6 | p < .001 | - |
| **Skin fluorescence$_{Log10}$ (n = 177)** |  |  |  |  |  |  |
| **Age, years** | 0.8 | p < .001 | 15.4 | 0.6 | p < .001 | 20.8 |
| **HbA$_{1c}$, mmol/mol** | 1.8 | p < .001 | 8.1 | 1.0 | p = .036 | - |
| **Smoking, pack years** | 0.3 | p = .023 | 2.8 | 0.3 | p = .009 | - |

Estimates indicate the percentage increase in fluorescence by a one unit increase in either age (years), HbA$_{1c}$ (mmol/mol) or smoking pack years (years).

**Table 4. Broad-sense heritability analyses of lens and skin fluorescence adjusted for age.**

| Model\coeff. | MZ corr. | DZ corr. | A | C | D | E | h² | AIC |
|---|---|---|---|---|---|---|---|---|
| **LF$_{log10}$ (54 MZ pairs; 50 DZ pairs)** | | | | | | | | |
| ACE | 0.85 [0.77–0.90] | 0.42 [0.39–0.46] | 0.85 [0.80–0.91] | 0.00 [0.00–0.00] | - | 0.15 [0.09–0.22] | 0.85 [0.78–0.91] | -338.78 ‡ |
| ADE | 0.85 [0.77–0.90] | 0.33 [0.05–0.57] | 0.49 [-0.57–1.55] | - | 0.36 [-0.70–1.42] | 0.15 [0.09–0.22] | 0.85 [0.78–0.91] | -339.26 ‡ |
| AE | 0.85 [0.77–0.90] | 0.42 [0.39–0.46] | 0.85 [0.78–0.91] | - | - | 0.16 [0.09–0.22] | 0.85 [0.78–0.91] | -340.78 † |
| CE | 0.64 [0.51–0.74] | 0.64 [0.51–0.74] | - | 0.64 [0.52–0.75] | - | 0.36 [0.25–0.48] | - | -314.52 |
| DE | 0.84 [0.77–0.90] | 0.21 [0.19–0.23] | - | - | 0.84 [0.78–0.91] | 0.16 [0.09–0.22] | 0.84 [0.78–0.91] | -340.58 ‡ |
| **SF$_{log10}$ (36 MZ pairs; 43 DZ pairs)** | | | | | | | | |
| ACE | 0.51 [0.28–0.69] | 0.26 [0.15–0.36] | 0.51 [0.30–0.72] | 0.00 [0.00–0.00] | - | 0.49 [0.28–0.70] | 0.41 [0.30–0.72] | -365.81 |
| ADE | 0.53 [0.30–0.70] | 0.15 [-0.16–0.43] | 0.07 [-1.16–1.29] | - | 0.46 [-0.80–1.72] | 0.47 [0.27–0.67] | 0.53 [0.33–0.73] | -366.25 ‡ |
| AE | 0.51 [0.28–0.69] | 0.26 [0.15–0.36] | 0.51 [0.30–0.72] | - | - | 0.49 [0.28–0.70] | 0.51 [0.30–0.72] | -367.81 ‡ |
| CE | 0.37 [0.17–0.55] | 0.47 [0.17–0.55] | - | 0.37 [0.18–0.57] | - | 0.63 [0.43–0.82] | - | -364.25 |
| DE | 0.53 [0.30–0.70] | 0.13 [0.08–0.18] | - | - | 0.53 [0.33–0.73] | 0.47 [0.27–0.67] | 0.53 [0.33–0.73] | -368.24 † |

Fluorescence parameters and age were statistically significant in all regression models (all p < .001). Heritability coefficient results denote the relative contribution to variance within each fluorescence parameter (i.e. 0.85 = 85%). **A:** Variance attributable to additive genetics, **ACE/ADE/AE/CE/DE**: Different combinatory heritability models. **AIC** = Akaike information criterion, **C:** Variance attributable to shared environment, **coeff.:** Coefficient, **corr.:** Correlation, **D:** Variance attributable to dominant genetics, **DZ:** Dizygotic, **E:** Variance attributable to non-shared environment, **h²**: Broad sense heritability (A + D), **LF:** Lens fluorescence, **MZ:** Monozygotic, **SF:** Skin fluorescence

†: Best fitting model within each tissue fluorescence type according to AIC criteria

‡: Statistically non-inferior models compared to best fitting model within each tissue fluorescence type according to AIC criteria.

lens fluorescence has consistently been found to increase with age and corresponding findings have been made for some skin fluorophores [11,12,20,37,46–50]. The variance in lens fluorescence explained by age in our univariate analysis ($R^2$ = 32%) was roughly comparable to that found by an earlier study ($R^2$ = 25%) of 59 subjects aged 8–91 years [38]. However, age explained considerably less of the variance in skin fluorescence in our study ($R^2$ = 15%) compared to that of another previous investigation ($R^2$ = 29%) conducted by van Waateringe et al. [42]. This discrepancy may in part be due to a much larger sample size or differences in demographics in the latter study (n = 8695, mean age 49 years ± SD 11) compared to our study population (n = 177, median age 59 (IQR 14)).

In heritability analyses, lens fluorescence was predominantly a heritable trait, whereas genetic and environmental influences were roughly balanced for skin fluorescence. It is presently unknown whether this notable difference in heritability depends on variations in glycosylation rates, factors that inhibit glycation or processes that lead to the degradation of glycation products in the two types of tissue examined by fluorescence in this study. The fact that we found only a crude association between fluorescence of the lens and skin also points to notable intrinsic differences in the accumulation and turnover of AGEs in the two tissues.

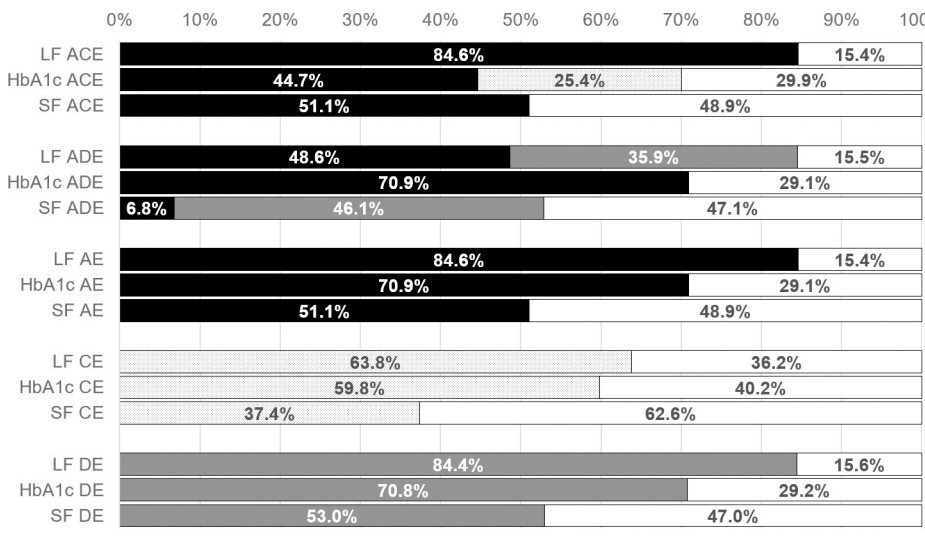

**Fig 3. Heritability coefficients of broad-sense heritability analyses adjusted for age.** Genetic factors (A and D) are displayed in dark color tones and environmental factors (C and E) are shown in bright colors. Broad-sense heritability (A + D = $h^2$) was highest for lens fluorescence across all models and lowest for skin fluorescence in 3/4 models. Analyses included the following number of subjects: $n_{\text{lens fluorescence}}$ = 208 (54 MZ and 50 DZ pairs), $n_{\text{skin fluorescence}}$ = 158 (36 MZ and 43 DZ pairs), $n_{\text{HbA1c}}$ = 318 (85 MZ and 74 DZ pairs). **A:** Additive genetics, **ACE/ADE/AE/CE/DE:** Different combinatory heritability models, **C:** Shared environment, **D:** Dominant genetics, **E:** Non-shared environment, **LF:** Lens fluorescence, **SF:** Skin fluorescence.

Being the parameter influenced the most by environmental factors, skin fluorescence could capture more of the variation in systemic AGE loads attributable to immediate lifestyle influences relative to fluorescence of the lens. The influences of accumulated, more long-term measures of life-style factors, however, may be better embedded in lens fluorescence, as we found the variance in fluorescence explained by life-time smoking pack years to be marginally higher for lens than skin fluorescence. Conversely, we found that current normoglycemia examined by $HbA_{1c}$ explained a higher proportion of the variance in skin fluorescence than in lens fluorescence. These findings suggest that skin fluorophores may have a shorter turnover than lens fluorophores, which may make fluorescence of the skin more indicative of recent systemic AGE loads compared to that of the lens.

The prominent heritability of lens fluorescence of 85% compared to the weaker heritability of $HbA_{1c}$ of 71% and type 2 diabetes of 20–80% [51] may explain why lens fluorometry is of limited value in screening for and monitoring of type 2 diabetes. It is likely that the reflection of very long-term AGE accumulation embedded in lens fluorescence is of limited relevance in assessing the current maintenance condition of diabetic individuals. Of the two types of tissue fluorescence presently examined, skin fluorescence was found to be more strongly associated with current glucose metabolism. Skin fluorometry was also less contaminated by genetics, which may be more practical in diabetes management, as development of type 2 diabetes is a product of complex interactions between not only genes but also, to a considerable degree, environmental factors [52]. Associations between skin fluorescence and a wide variety of type 2 diabetes markers and long-term diabetes complications have been reported [53]. Assessing risks of diabetes complications is arguably of higher clinical interest than simply diagnosing diabetes [54]. For this purpose, skin fluorescence may potentially supplement $Hab_{1c}$ as a biomarker in diabetes management.

## Strengths and limitations

Strengths of the study include the twin design and, for a twin study, the number of participants. To our knowledge, this is presently the largest comparative study of lens and skin fluorescence and the first to compare the heritability of lens fluorescence and skin fluorescence. Limitations include that smoking was quantitated solely based on interview so that data may have been subjected to interviewer or recall biases. Ideally, broad-sense heritability analyses should be performed on groups consisting of only men or only women, but we choose to include both sexes to increase statistical power.

## Supporting information

**S1 File. Supplementary materials.**
(DOCX)

## Acknowledgments

Clinical Research Associate Professor, MD, Ph.D, Line Kessel at the Department of Ophthalmology, Rigshospitalet Glostrup and president of Ocumetrics Bruce M. Ishimoto were helpful with practical guidance regarding lens fluorometry. MD Mustafa Al-Hamdani contributed with data collection.

## Author Contributions

**Conceptualization:** Jakob Bjerager, Mohamed Belmouhand, Simon P. Rothenbuehler, Birgit Sander, Michael Larsen.

**Data curation:** Jakob Bjerager, Sami Dabbah, Mohamed Belmouhand.

**Formal analysis:** Jakob Bjerager, Birgit Sander, Michael Larsen.

**Funding acquisition:** Jakob Bjerager, Mohamed Belmouhand, Simon P. Rothenbuehler, Birgit Sander, Michael Larsen.

**Investigation:** Jakob Bjerager, Simon P. Rothenbuehler, Birgit Sander, Michael Larsen.

**Methodology:** Jakob Bjerager, Simon P. Rothenbuehler, Birgit Sander, Michael Larsen.

**Project administration:** Jakob Bjerager, Sami Dabbah, Mohamed Belmouhand, Simon P. Rothenbuehler, Michael Larsen.

**Resources:** Jakob Bjerager, Sami Dabbah, Mohamed Belmouhand, Simon P. Rothenbuehler, Michael Larsen.

**Software:** Jakob Bjerager, Birgit Sander, Michael Larsen.

**Supervision:** Jakob Bjerager, Sami Dabbah, Mohamed Belmouhand, Birgit Sander, Michael Larsen.

**Validation:** Jakob Bjerager, Birgit Sander, Michael Larsen.

**Visualization:** Jakob Bjerager, Birgit Sander, Michael Larsen.

**Writing – original draft:** Jakob Bjerager, Birgit Sander, Michael Larsen.

**Writing – review & editing:** Jakob Bjerager, Sami Dabbah, Mohamed Belmouhand, Birgit Sander, Michael Larsen.

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
