## [Decision Letter · Decision Letter 0]

4 Aug 2021

PONE-D-21-09313

Lens fluorescence and skin fluorescence in the Copenhagen Twin Cohort Eye Study: Covariates and heritability

PLOS ONE

Dear Dr. Bjerager,

Thank you for submitting your manuscript to PLOS ONE. After careful consideration, we feel that it has merit but does not fully meet PLOS ONE’s publication criteria as it currently stands. Therefore, we invite you to submit a revised version of the manuscript that addresses the points raised during the review process.

We look forward to receiving your revised manuscript.

Kind regards,

Ayse Ulgen, PhD, MGM

Academic Editor

PLOS ONE

Journal Requirements:

This work was supported by THE VELUX FOUNDATIONS (JB, grant no. 00028975, https://veluxfoundations.dk/en), Rigshospitalets Forskningsudvalg (MB, grant no. E-23334-02, https://www.rigshospitalet.dk/forskning/om-forskningen/Sider/forskningsudvalget.aspx), P. Carl Petersens Fond (MB, grant no. 19102, https://www.pcarlp-fond.dk/), Helsefonden (MB, grant no. 19-B-0063, https://helsefonden.dk/), Aase og Ejnar Danielsens Fond (MB, grant no. 18-10-0698, https://danielsensfond.dk/), Beckett Fonden (MB, grant no. 19-2-3490, https://beckett-fonden.dk/) and Einar Willumsens Mindelegat (MB, grant no. 500028, https://www.legatbogen.dk/fabrikant-einar-willumsens-mindelegat/stoetteomraade/7684).

Reviewers' comments:

Reviewer's Responses to Questions

**Comments to the Author**

1. Is the manuscript technically sound, and do the data support the conclusions?

Reviewer #1: Partly

Reviewer #2: No

Reviewer #3: Yes

2. Has the statistical analysis been performed appropriately and rigorously? 

Reviewer #1: Yes

Reviewer #2: Yes

Reviewer #3: Yes

3. Have the authors made all data underlying the findings in their manuscript fully available?

Reviewer #1: Yes

Reviewer #2: Yes

Reviewer #3: Yes

4. Is the manuscript presented in an intelligible fashion and written in standard English?

Reviewer #1: No

Reviewer #2: Yes

Reviewer #3: Yes

5. Review Comments to the Author

Reviewer #1: General comments

The authors examined lens and skin fluorescence and compared to well-described risk factors (age, glucose, smoking). The value of these measurements is not clear from the manuscript and it does not seem that much new information is added to what has previously been described. Thus, as a reader I´m not sure why I should read this manuscript and what I (or the scientific community) learn from the study.

The main problem with the manuscript is that it is not explained why anyone would want to read about lens and skin fluorescence. It seems the main reason has something to do with diabetes - but then why are all participants with diabetes excluded from the study?

The authors seem to argue that skin and lens fluorescence are two sides of the same issue - but if this is so - why is the correlation between skin and lens fluorescence so low? If they measure the same thing one would expect them to show a stronger correlation. If they do not measure the same thing - why mix them in this manuscript?

The manuscript is extremely long and should be significantly shortened, e.g. the Results section uses a whole page just to describe the study population before any results are presented.

Specific comments

Abstract: there seems to be something wrong with the IQR of age?

Statistical methods: the authors seem to have relied primarily on correlation coefficients, however the estimate of correlation coefficients depends not only on the strenght of the association between variables but also on the number of observations. I would recommend to use more advanced methods.

Data availability: I only get an error message when I copy the web-adress?

Introduction: the Introduction section is much too long. I would recommend to reduce by at least 50% and to keep a focus on why the reader should be interested in the lens and skin fluorescence.

Language: there are several strange things in the manuscript, e.g. line 120 "intraocular tonometry" - did the authors measure intraocularly? and line 120 continued - cognitive impairment that contravened informed consent - what does contravene mean in this sentence?

Ethical approval: it is stated that the local ethical committee gave approval but they refer to a national committee?

Page 5 "Three successful scans were attempted for each subject" if three successful scans were not obtained - did they try a fourth? or fifth?

Results: the description of who were included is very lengthy and basically repeats exclusion criteria - I would recommend a flow chart rather than the many words.

Reproducibility: I don´t understand this section. Do the authors try to convince the reader that the measurements are reliable? I would recommend to move to supplementary materials rather than in the main text.

The name of the cohort study does not need to be repeated so many times - once is enough.

Who was examined? the authors state a number of reasons that not all were examined but these reasons seems rather strange, "reprioritization of examination modalities"? The main problem with the manuscript and its content is the lack of applicability of methods and this sentence makes me think not even the authors decided the examinations were important? then why should anybody want to read about them?

Discussion: I would strongly recommend not to start by repeating the Introduction but rather get to the point of what the reader should learn from this manuscript.

Line 290 onwards: maybe the reason the authors did not find any convincing associations with glucose levels was because they had omitted all participants who were not normoglycemic which I assume means that the variation in glycemic load in the population would probably have been too small to be detected.

Table 6: why report the dindings from a different study so detailed? I would recommend to remove Table 6. 1½ page is used to describe the differences between this and a similar previous study. This could be significantly shortened. Readers with a great interest in skin and lens fluorescence are likely to already know the previous manuscript and to compare the two on his/her own hand.

line 356: " We found skin fluorescence to provide a more up-to-date estimate of normal glucose metabolism than lens fluorescence." NO, you did not find this, you did not even examine this, then you shouldn´t write that you found it

Line 365-366 "We encourage future research to investigate if skin fluorometry is more relevant than lens fluorometry in discerning the diabetic phenotype from normality " Why encourage this? why not simply rely on blod glucose measurements rather than a proxy measure that you yourself find to be a poor proxy measure? this seems absurd

Limitations: it is mentioned that it is a limitation of the study that several authors performed the measurements - but does this mean that some examiners could not obtain reliable measurements? is this why so much attention is paid to the validity of measurements?

Reviewer #2: The authors in the manuscript investigated fluorescence levels in lens and skin. They draw conclusion, skin fluorescence is preferable over lens fluorescence in predicting estimate of normo-glycemia. Following are the concerns:

1. Are the methods used to measure fluorescence of lens and skin reported previously? If so, please cite references.

2. The authors in the manuscript are biased towards fluorescence, that they observe from skin or lens, are due to glycation. No experimental evidence is provided for such claims in the manuscript. Please provide supporting evidences for such claims in the manuscript. Please note, fluorescence in lens or skin could be also due to many other post-translational modifications, like oxidation.

3. The authors used linear regression fit to corelate fluorescence levels in lens and skin with age. Is similar trend found in other previous reports? Please cite evidences.

4. Please add controls for measuring fluorescence in lens and skin.

5. In Fig 1, plot of skin and lens fluorescence is highly scattered. It is hard drawing any correlation. This is the reason for low R2 value. Thus, conclusions drawn from this are doubtful.

Reviewer #3: The author examined lens fluorescence and skin fluorescence in 163 effective samples, and studied the interrelatedness between fluorescence parameters and relations between 50 fluorescence and age, current HbA1c and smoking pack years. They established the variance explained in lens fluorescence by the variables. Also, they showed that the age-adjusted heritability for lens fluorescence, fluorescence and HbA1c. And reached the conclusion that lens fluorescence was more predominantly heritable, whereas skin fluorescence was more influenced by environmental factors and closer related to current glycemia. Thus, skin fluorophores have a faster turn-over than lens fluorophores and that skin fluorescence reflects a more up-to-date estimate of current normo-glycemia.

The statistical analysis is sounds and the manuscript is clear written.

6. PLOS authors have the option to publish the peer review history of their article (what does this mean?). If published, this will include your full peer review and any attached files.

Reviewer #1: No

Reviewer #2: No

Reviewer #3: No

---

## [Author Response · Author response to Decision Letter 0]

14 Aug 2021

Rebuttal letter regarding PLOS ONE manuscript submission: 

‘Lens fluorescence and skin fluorescence in the Copenhagen Twin Cohort Eye Study: Covariates and heritability’

PLOS ONE [PONE-D-21-09313] - [EMID:2d20a953098b23dc]

We have addressed all points raised during the review process, which we go through below: 

Regarding Journal requirements: 

1. “Please ensure that your manuscript meets PLOS ONE's style requirements, including those for file naming (…)”

The manuscript has now been fully reformatted according to the PLOS ONE Manuscript Body Formatting Guidelines, the PLOS ONE Title, Author, Affiliations Formatting Guidelines. 

2. “Please state what role the funders took in the study. (…)”

The founders took no role in the study. The following sentence has been added to the Funding section: "The funders had no role in study design, data collection and analysis, decision to publish, or preparation of the manuscript." 

3. “(…) Should your manuscript be accepted for publication, we will hold it until you provide the relevant accession numbers or DOIs necessary to access your data. (…)”

Please excuse us for having provided a link to our data repository that will first become active after publication. The correct URL for previewing the dataset is:

https://datadryad.org/stash/share/4mKR7uv3YYWkcWgLvTN9hMJM5EUS5Wxm9grPMxbvMpE

4. “We note that you have included the phrase “data not shown” in your manuscript. Unfortunately, this does not meet our data sharing requirements. (…)”

The first occurrence of the phrase “data not shown” in line 258 regarding the best fitting heritability model for HbA1c has been replaced with data (AIC values for the heritability models mentioned). 

The second occurrence of the phrase in line 151 has been removed along with its associated sentence, which stated that sex had no impact on fluorescence parameters in explorative analysis. 

Regarding Reviewer #1: 

General comments

"The authors examined lens and skin fluorescence and compared to well-described risk factors (age, glucose, smoking). The value of these measurements is not clear from the manuscript and it does not seem that much new information is added to what has previously been described. Thus, as a reader I´m not sure why I should read this manuscript and what I (or the scientific community) learn from the study."

We thank the reviewer for these points, which we hope to clarify. 

Numerous studies have investigated the diagnostic potential of lens and skin fluorescence in diabetes screening and risk assessment for diabetes complications (references 11,12,21–25,13–20). Although the diagnostic potential of lens fluorescence in diabetes screening is continuingly researched (Sertbas et al. 2019; doi: 10.20452/pamw.4449, Pehlivanoğlu et al. 2018. doi: 10.2147/OPTH.S164960. Cahn et al. 2014; doi: 10.1177/1932296813516955), the scientific interest in this topic is arguably declining. There is, however, a considerable and growing research output regarding skin fluometry with 194 papers published as of 2020 (Source: https://www.diagnoptics.com/wp-content/uploads/2021/07/AGE-Reader-Publication-List-05172021.pdf), most notably in relation to morbidity and mortality in diabetes, cardiovascular disease and the metabolic syndrome.

Only one, small-scale study has previously compared lens and skin fluorescence (Januszewski et al. 2012). Comparisons between the modalities are relevant in order to elucidate strengths and limitations to the methods, as there is an overlap of which fluorescent AGEs accumulate in both tissues (references 4-8). To our knowledge, we have conducted the largest study on lens and skin fluorescence and the first comparative heritability study of the two parameters. We add to the literature that lens fluorescence is highly heritable while the variance in skin fluorescence is equally balanced by genetic and environmental factors. Also, of lens and skin fluorescence, the latter is more strongly associated with current glycemia. Our findings help explain the large proportion of residual variance in lens and skin fluorescence, which is presently not accounted for by known covariates. This may have implications for choices of future researchers, as we find that skin fluorescence is probably the more promising modality to investigate further, although effect sizes are small. In addition, ageing processes is generally an important research topic for the future. Our findings contribute to the understanding of ageing kinetics in lens and skin tissues, reflected by AGE accumulation. 

We have now streamlined the manuscript on the basis of all points raised in the review process, and we hope that this reviewer now finds our aims to be more concisely delivered (e.g. line 62-65 and 271-298). Please let us know if the reviewer has further inquiries. 

"The main problem with the manuscript is that it is not explained why anyone would want to read about lens and skin fluorescence. It seems the main reason has something to do with diabetes - but then why are all participants with diabetes excluded from the study?"

We have now tried to clarify why this topic of research is relevant (lines 62-65):

“Research into the normal variation of tissue fluorescence parameters may broaden our understanding of tissue ageing kinetics as well as assist in evaluating the clinical potential of these methods in management of diabetes, the metabolic syndrome (MetS) and cardiovascular disease, all of which are related to increased tissue fluorescence [12,13,23,41,43,44].”

We have compared two quantitative modalities that are thought to reflect glycemic stress loads and risk of metabolic disease and complications. We believe that our contribution to the literature may help guide future researchers within the fields of ageing, metabolic disease and diagnostics decide on which direction to follow next. As mentioned in response to the reviewer’s precious comment, there has been a recent surge in research output regarding skin fluorometry. We believe that there is a scientific audience that is interested in further knowledge into the normal variation and confounding factors that affect tissue fluorescence, and that put the characteristics of skin fluorescence into perspective. 

On why subjects with diabetes were omitted, please see our response to a later specific comment by this reviewer (“Line 290 onwards: maybe the reason the authors did not find any convincing (…)”)

"The authors seem to argue that skin and lens fluorescence are two sides of the same issue - but if this is so - why is the correlation between skin and lens fluorescence so low? If they measure the same thing one would expect them to show a stronger correlation. If they do not measure the same thing - why mix them in this manuscript?"

We agree that a correlation of R = 0.38 is modest at best. This is interesting, considering the intuitive hypothesis that AGE formation in both tissue types share common genesis and biochemical pathways. We believe that the low correlation points to marked differences in cell-turnover and/or kinetics of glycosylation rates in the two tissues, which we found support for in heritability analyses. We elaborate on this in the last half of the discussion. In the end we agree with reviewer that the two fluorescence modalities should probably not be mixed (lines 285-295). 

In addition, the previous small-scale study comparing lens and skin fluorescence (Januszewski et al. 2012) found a markedly higher correlation between lens and skin fluorescence. We believe that our finding of a cruder correlation is important as to moderate and supplement earlier findings. 

"The manuscript is extremely long and should be significantly shortened, e.g. the Results section uses a whole page just to describe the study population before any results are presented."

We have now significantly shortened the manuscript, primarily in the introduction, results and discussion sections, on the basis of all points raised by reviewers. We hope the reviewer is satisfied with the new appearance of the results section. 

Specific comments

"Abstract: there seems to be something wrong with the IQR of age?"

The abstract states that the median age of included subjects was 59 (IQR 11-14). 

The IQR interval covers that the median age was 59 (IQR 11), 59 (IQR 14) and 59 (IQR 13) for the lens fluorescence population, skin fluorescence population and the population in which we had both lens and skin fluorescence measurements, respectively (Table 1). Our reason not to state a combined median age with a single IQR number for subjects across all subgroups is that the number does not appear elsewhere in the article, in which case the abstract would present unique data. No analysis combines all individuals of all subgroups. 

We have now removed the indication of IQR(s) from the abstract to avoid confusion for readers. 

"Statistical methods: the authors seem to have relied primarily on correlation coefficients, however the estimate of correlation coefficients depends not only on the strenght of the association between variables but also on the number of observations. I would recommend to use more advanced methods."

We agree with the limitations to correlations coefficients pointed to by the reviewer. We choose to include them as they are the only means by which we can compare our findings to the only previous study comparing aspects of lens and skin fluorescence (Januszewski et al. 2021) . Our number of observations were 3-4-fold higher than those in the study Januszewski et al., and still we found markedly lower correlations between all study parameters. We believe this is a relevant contribution to the literature, as it emphasizes a strong effect of increasing age on the inter-individual variation in fluorescence parameters groups (our study population (median age 59 years IQR 11-14) was substantially older that of the former study (mean age 36 years ± SD 13)).

To reduce the emphasis on correlation testing, results of these analyses have been largely removed from the main text so that they now mostly stand alone in Table 2. To supplement with a more advanced analysis, we have included an adjusted R2 coefficient from linear regression mixed model analysis adjusted for twin pair clustering for comparison to the correlation test between lens and skin fluorescence (lines 177-179).

In our revised manuscript, we find that the discussion section now mainly emphasizes the implications of our more sophisticated analyses (linear mixed model regression analyses adjusted for twin data clustering of fluorescence parameters as a function of covariates and linear regression broad-sense heritability analyses of fluorescence parameters and HbA1c adjusted for age) rather than the results of correlation testing. We hope the reviewer also finds that this is the case. 

"Data availability: I only get an error message when I copy the web-adress?"

We had by mistake posted a link that will first become active when the article and dataset have been published. Please excuse us the inconvenience. The correct link for previewing the dataset before publication by the publisher Dryad is posted below: 

https://datadryad.org/stash/share/4mKR7uv3YYWkcWgLvTN9hMJM5EUS5Wxm9grPMxbvMpE

"Introduction: the Introduction section is much too long. I would recommend to reduce by at least 50% and to keep a focus on why the reader should be interested in the lens and skin fluorescence."

The introduction has been focused and reduced by 50%. We hope that the reviewer is pleased with the new version. 

"Language: there are several strange things in the manuscript, e.g. line 120 "intraocular tonometry" - did the authors measure intraocularly? and line 120 continued - cognitive impairment that contravened informed consent - what does contravene mean in this sentence."

“Intraocular (previously line 120, now line 76) has been corrected to “ocular”, please excuse us for the typing error. “Contravene” has been replaced with “hindered”. 

"Ethical approval: it is stated that the local ethical committee gave approval but they refer to a national committee?"

We have now corrected “local” to “national” (line 80). Due to single municipality ethics committee in Denmark we used “local” interchangeably with “national”, but we agree that “national” is the correct statement. 

"Page 5 "Three successful scans were attempted for each subject" if three successful scans were not obtained - did they try a fourth? or fifth?"

Correct, up to six scans were attempted in cases were three successful scans were not achieved by the first three attempts. We have rephrased the part to clarify this: 

“Subjects were scanned up to six times in order to achieve three successful scans. Individuals were excluded from the study if six scan attempts could not provide three successful scans.”

(lines 100-102)

"Results: the description of who were included is very lengthy and basically repeats exclusion criteria - I would recommend a flow chart rather than the many words."

A flow chart has been added to replace much of the description. The text still describe the population in which both lens and skin fluorescence measurements were available, and the population used for heritability analysis of HbA1c, as this information is not listed elsewhere. 

"Reproducibility: I don´t understand this section. Do the authors try to convince the reader that the measurements are reliable? I would recommend to move to supplementary materials rather than in the main text."

We thank the reviewer for helping us clarify this topic. 

We included data on repeatability to provide a measure of the inter-scan precision of the methods. The considerable residual variance in fluorescence parameters that age, smoking pack years and current HbA1c could not explain could in part have been due to imprecision of the fluorometric methods, if the repeatability had been low. The fact that we found a high repeatability of both methods suggest that the residual variance is more likely attributable to genetics or covariates not accounted for in this study (and the literature).

We have added the following explanation to the manuscript: 

“As our fluorescence measurements were highly repeatable with no statistically significant difference between measurements of individuals, the considerable residual variance in fluorescence parameters left unexplained cannot be meaningfully attributed to a lack of precision in the fluorometric methods used, and it is likely better explained by either genetical influences or environmental factors unaccounted for in the present study.” (lines 238-242).

Also, we have changed the wording from “reproducibility” to “repeatability”, which is the correct term in this context. The results of repeatability measurements have also been moved to supplementary materials (Supplementary Materials 1 or S1)) according to the suggestion by the reviewer. 

"The name of the cohort study does not need to be repeated so many times - once is enough."

We agree, repetitions after first mentioning have been removed. 

"Who was examined? the authors state a number of reasons that not all were examined but these reasons seems rather strange, "reprioritization of examination modalities"? The main problem with the manuscript and its content is the lack of applicability of methods and this sentence makes me think not even the authors decided the examinations were important? then why should anybody want to read about them?"

The cohort study commenced in February 2019. Unfortunately, the arrival of our skin fluorescence device (Diagnoptics AGE Reader) was markedly delayed by the manufacturer, which hindered acquisition of skin fluorescence readings until June 2019. Lens fluorescence measurements had been performed up until this point and continued. The skin fluorescence reader was leased to us on a contract that expired in early January 2020, which hindered further skin fluorescence measurements. At this point we decided to also stop conducting lens fluorescence measurements as well, as our goal was to compare lens to skin fluorescence as well as perform a 21-year follow-up study on lens fluorescence, and all follow-up participants except for 3 individuals had been reexamined by January 2020. 

Our study was part of a large cohort study with an extensive 3-hour eye examination protocol and the study had numerous collaborators and research interests. Stopping further lens fluorescence measurements at the time when skin fluorescence could also not be performed anymore freed up valuable minutes to perform new eye imaging modalities. 

We have replaced the original phrasing with the more accurate description stated below:

 “Lens and skin fluorometry was planned and attempted in all individuals who were examined in 2019, limitations being late instrument delivery and expiring instrument lease contract of the skin fluorometer.” (lines 82-84). 

Regarding why this article has relevance, please see our responses to the first two general comments by reviewer 1. 

"Discussion: I would strongly recommend not to start by repeating the Introduction but rather get to the point of what the reader should learn from this manuscript."

We have rearranged the paragraphs of the discussion to align with the wishes of the reviewer. The original opening paragraph of the discussion has been shortened, split up and moved to other parts of the discussion. (now lines 243-250 and lines 272-274). 

"Line 290 onwards: maybe the reason the authors did not find any convincing associations with glucose levels was because they had omitted all participants who were not normoglycemic which I assume means that the variation in glycemic load in the population would probably have been too small to be detected."

We agree with the reviewer. If possible we would have analyzed subjects with diabetes as a subgroup, but unfortunately the low prevalence of diabetic subjects in our study population (half the rate of the background population) did not provide a meaningful basis for doing so. Normal distributions of the duration of diabetes and the maintenance condition of individuals would not possibly be attainable. Adjustment for treatment with antidiabetics could not have been done meaningfully. 

Including the few participants with diabetes in the present study would create new problems, as they prove to be strong outliers both concerning fluorescence and HbA1c. Comparison of results with those of earlier studies would also prove less ideal, as these studies mostly discern between groups of subjects with diabetes as opposed to non-diabetics (healthy). 

Due to these reasons, we decided to exclude subjects with diabetes and make a focused study of the normal variation of parameters in healthy individuals. Due to our large sample sizes we expected to still be able to detect a meaningful variation in associations to HbA1c, which we also did regarding the influence of HbA1c on skin fluorescence (p< .001 in univariate and p = .036 in multivariate linear regression analyses).

We have made the following addition to the methods section:

“Subjects with diabetes were excluded due to strong outlier effects. The prevalence of subjects with diabetes was too low to allow for meaningful comparisons between subjects with and without diabetes.” (lines 84-86).

"Table 6: why report the dindings from a different study so detailed? I would recommend to remove Table 6. 1½ page is used to describe the differences between this and a similar previous study. This could be significantly shortened. Readers with a great interest in skin and lens fluorescence are likely to already know the previous manuscript and to compare the two on his/her own hand."

Table 6 and most comparisons to the study by Januszewski et al. has been deleted, including the elaboration of methodological discrepancies between the present and the former study. 

"line 356: " We found skin fluorescence to provide a more up-to-date estimate of normal glucose metabolism than lens fluorescence." NO, you did not find this, you did not even examine this, then you shouldn´t write that you found it"

We have rephrased our statement to the following: 

“Of the two types of tissue fluorescence presently examined, skin fluorescence was found to be more strongly associated with current glucose metabolism.” (lines 288-290)

Also, a similar notion in the abstract have been rephrased accordingly. (line 40)

"Line 365-366 "We encourage future research to investigate if skin fluorometry is more relevant than lens fluorometry in discerning the diabetic phenotype from normality " Why encourage this? why not simply rely on blod glucose measurements rather than a proxy measure that you yourself find to be a poor proxy measure? this seems absurd"

We absolutely agree that HbA1c will remain the cornerstone of diabetes diagnosis and management, but as we have stated in the introduction, HbA1c only explains marginal amounts of risks of developing diabetes complications (e.g. 11% of the risk for developing diabetic retinopathy). Biomarkers that can potentially support HbA1c in explaining higher risk fractions may prove clinically useful, which is in line with the recently considerable research interest in skin fluorescence and associations with type 2 diabetes parameters mentioned previously. 

We have revised the paragraph to the following:

“Assessing risks of diabetes complications is arguably of higher clinical interest than simply diagnosing diabetes [52]. For this purpose, skin fluorescence could potentially supplement Hab1c as a biomarker in diabetes management.” (lines 294-297)

"Limitations: it is mentioned that it is a limitation of the study that several authors performed the measurements - but does this mean that some examiners could not obtain reliable measurements? is this why so much attention is paid to the validity of measurements?"

We are confident that all examiners could obtain reliable measurements, but lens fluorometry with the Ocumetrics FluorotronMaster is a semi-manual procedure. Although we do not expect results to have been influenced by operator bias, we cannot exclude it, and therefore we would rather disclaim it to the reviewing committee than being coitized for omitting this information. 

We have now removed the statement from the strengths and limitations section. 

Reviewer #2: 

"1. Are the methods used to measure fluorescence of lens and skin reported previously? If so, please cite references."

Yes, the lens fluorometer (Ocumetrics FluorotronMaster) and skin fluorometer (Diagnoptics AGE Reader) have been used in most studies regarding lens and skin fluorescence, of which we numerous in the introduction (references [11,12,21–25,13–20] in lines 49 and [12,13,23,41,43,44] in line 65).

"2. The authors in the manuscript are biased towards fluorescence, that they observe from skin or lens, are due to glycation. No experimental evidence is provided for such claims in the manuscript. Please provide supporting evidences for such claims in the manuscript. Please note, fluorescence in lens or skin could be also due to many other post-translational modifications, like oxidation."

We thank the reviewer for this consideration. 

It is true that there are numerous types of post-translational modification of proteins, many of which may contribute to tissue fluorescence. However, studies have found the concentration of AGEs extracted from tissue samples to account for the meaningful part of the fluorescence measured in lens and skin tissue samples (Mota et al. 1994, Genuth et al. 2005, Meerwaldt et al. 2004, Monnier et al. 1999, Kessel et al. 2002). In the lens, only marginal amounts of fluorescence can be attributed to kyneurines, which are the products of photooxidized tryptophane, and its concentration seems to actually decrease with age (Kessel et al 2002). 

We have cited the abovementioned studies in the introductory paragraph (line 48), but we do not elaborate further on this topic since reviewer 1 requested that we shortened the introduction by 50 %. We hope that reviewer 2 is satisfied with this course of action. 

"3. The authors used linear regression fit to corelate fluorescence levels in lens and skin with age. Is similar trend found in other previous reports? Please cite evidences."

Yes, we reported comparisons and references to previous studies regarding lens and skin fluorescence with age (Van Best et al. 1998 and van Waateringe et al 2016), (lines 263 and 267) 

"4. Please add controls for measuring fluorescence in lens and skin."

Our study was explorative in nature and conducted on volunteering, non-diabetic individuals who were in self-reported good health and recruited from a population-based registry. We believe that the exclusion of diabetic individuals qualifies for an inquiry into the normal range and variation of lens and skin fluorescence, as diabetes significantly increase tissue fluorescence levels.

The only previous study comparing lens and skin fluorescence (Januszewski et al. 2021) compared 69 subjects with diabetes to 60 controls (“Control subjects had no current or past history of diabetes, no known cardiovascular or renal disease and had normal fasting glucose and renal function.”) The characteristics of our total study population effectively corresponds to the control group in the former study. As such, we do not see that further criteria could be used to meaningfully differentiate a control group for comparison with our study population. 

We hope that the reviewer is satisfied with our response to this inquiry. 

"5. In Fig 1, plot of skin and lens fluorescence is highly scattered. It is hard drawing any correlation. This is the reason for low R2 value. Thus, conclusions drawn from this are doubtful."

We completely agree with the reviewer regarding the plot and that the low R and R2 values of the relationship between lens and skin fluorescence should be used to back positive claims. The fact that our sample size (n = 163) is presumably sufficiently large for meaningful correlation analyses suggest that the low R and R2 values point to an absence of a meaningful relationship. 

We have stated this in the discussion section:

“The fact that we only found a crude association between fluorescence of the lens and skin also points to notable intrinsic differences in the accumulation and turnover of AGEs in the two tissues” (lines 272-274)

Reviewer #3: 

"The statistical analysis is sounds and the manuscript is clear written."

We thank the reviewer for the time taken to perform this review.

---

## [Editor Report · Decision Letter 1]

20 Aug 2021

Lens fluorescence and skin fluorescence in the Copenhagen Twin Cohort Eye Study: Covariates and heritability

PONE-D-21-09313R1

Dear Dr. Bjerager,

We’re pleased to inform you that your manuscript has been judged scientifically suitable for publication and will be formally accepted for publication once it meets all outstanding technical requirements.

Kind regards,

Ayse Ulgen, PhD, MGM

Academic Editor

PLOS ONE

---

## [Editor Report · Acceptance letter]

31 Aug 2021

PONE-D-21-09313R1 

Lens fluorescence and skin fluorescence in the Copenhagen Twin Cohort Eye Study: Covariates and heritability 

Dear Dr. Bjerager:

I'm pleased to inform you that your manuscript has been deemed suitable for publication in PLOS ONE. Congratulations! Your manuscript is now with our production department. 

Kind regards, 

on behalf of

Dr. Ayse Ulgen 

Academic Editor

PLOS ONE